# DynaSTy: A Framework for SpatioTemporal Node Attribute Prediction in Dynamic Graphs

## Abstract

Accurate multi-step forecasting of node-level attributes on dynamic graphs is critical for applications ranging from financial trust networks to biological networks. Existing spatio-temporal graph neural networks typically assume a static adjacency matrix. In this work, we propose an end-to-end *dynamic edge-biased spatio-temporal model* that ingests a multidimensional time series of node attributes and a time series of adjacency matrices, to predict multiple future steps of node attributes. At each time step, our transformer-based model injects the given adjacency as an adaptable attention bias, allowing the model to focus on relevant neighbors as the graph evolves. We further deploy a masked node/time pretraining objective that primes the encoder to reconstruct missing features, and train with scheduled sampling and a horizon-weighted loss to mitigate compounding error over long horizons. Unlike prior work, our model accommodates dynamic graphs that vary across input samples, enabling forecasting in multi-system settings such as brain networks across different subjects, financial systems in different contexts, or evolving social systems. Empirical results demonstrate that our method consistently outperforms strong baselines on Root Mean Squared Error (RMSE) and Mean Absolute Error (MAE).

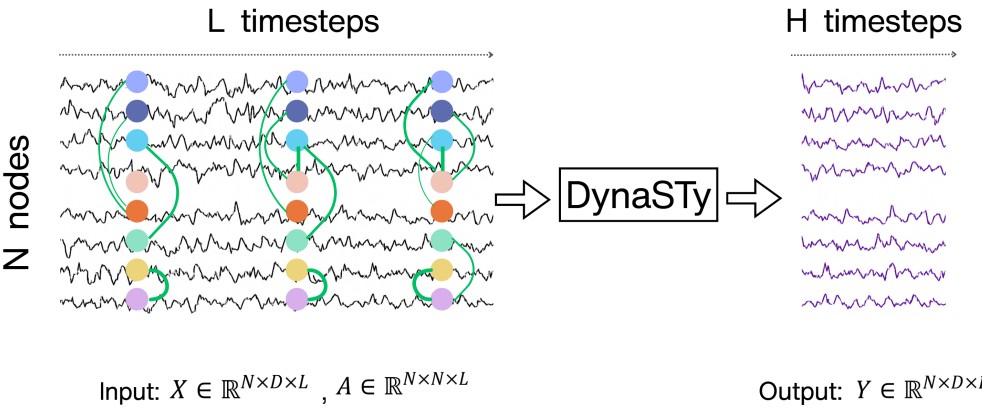

Figure 1: Overview of expected input and output of DynaSTy

## 1 Introduction

Many real-world systems, ranging from brain connectivity networks to social trust platforms, are naturally represented as dynamic graphs, where the set of edges and node attributes evolve over time. In these settings, the underlying relationships between entities (i.e., the graph structure) change due to external stimuli or internal dynamics. For example, in functional brain networks, edges correspond to time-varying functional connections between brain regions, which reconfigure dynamically in response to cognitive states or external tasks. In Bitcoin-OTC and Alpha trust networks, the trust rating interactions between users evolve as a result of transactions, leading to changing connectivity

over time. Similarly, in dynamic social or biological systems, interactions are not only sparse but also transient, making edge evolution a critical modeling component. While considerable progress has been made in learning from dynamic graphs, much of this work focuses on node classification (Sun et al. (2022); Ma et al. (2025); Song et al. (2025)) or link prediction (Mei & Zhao (2024); Tian et al. (2024)), often assuming fixed node connectivity. In contrast, predicting future node attributes, such as a node's behavioral signal, risk score, or physiological state, is both important and underexplored, especially in the presence of dynamic edge structures.

Most existing Spatiotemporal Graph Neural Network (STGNN) models, including DCRNN (Li et al. (2018)), STGCN (Yu et al. (2018)), and MTGNN (Wu et al. (2020)), assume a static input graph that remains fixed across all time steps and training examples. This design inherently restricts these models to learning from a single system of entities (e.g., a single traffic network), where both the node set and the relational structure are shared globally. However, in many real-world applications, such as brain network analysis, multi-subject behavioral tracking, or ecological monitoring, we are presented with multiple distinct systems that share a common node ontology (e.g., brain regions) but exhibit different relational dynamics. In these settings, each input example corresponds to a different spatiotemporal graph sequence.

We propose a node attribute prediction method, *DynaSTy*, that leverages the structure of the dynamic graphs while maintaining a fixed node set across samples. The model relaxes the assumption of a static global graph shared across training samples and allows a different dynamic graph per training sample. This makes the model directly applicable to domains such as fMRI-based brain region BOLD signal (Ogawa et al. (1990)) forecasting, where each subject has a different dynamic brain connectivity profile but shares the same set of anatomical regions. By modeling both temporal dynamics and sample-specific graph structure, our approach generalizes STGNNs to broader domains where individual graphs evolve differently across examples.

To evaluate our method, we consider semi-synthetic and real-world datasets including the LA traffic network and Bitcoin-OTC and Alpha trust networks, where node features represent things like traffic volume at an intersection or average trust ratings given and received. We also compare against strong baselines like STGCN (Yu et al. (2018)), DGCRN and MTGNN, showing that our model achieves superior performance on mean squared error and mean absolute error metrics on most datasets. To our knowledge, prior work rarely evaluates node-attribute forecasting under fully evolving edges, especially in the per-sample dynamic-graph setting; we explicitly target this regime.

## 2 RELATED WORK

**Dynamic Graph Representation Learning.** A large body of work has been devoted to learning on dynamic graphs, primarily targeting tasks such as link prediction and node classification. Methods such as EvolveGCN ( Pareja et al. (2019)), TGAT ( Dai et al. (2024)), TGN ( Rossi et al. (2020)), and DyRep ( Trivedi et al. (2018)) model temporal interactions in graphs by evolving either node embeddings or graph parameters over time. However, these methods typically focus on classification or event prediction and do not address the task of predicting continuous-valued node attributes. Furthermore, many prior works model graphs as streams of discrete events (e.g., interactions between node pairs), rather than explicitly modeling evolving graph snapshots with dense temporal node attributes. Event-stream models are flexible but can't always handle rich node attribute time series directly (e.g., vectors of features at each time step). Our method is better suited for settings where both topology and node attributes evolve continuously and are available at regular intervals.

**Time Series Forecasting with GNNs.** Several methods have explored forecasting node values in spatiotemporal settings, especially in traffic and sensor networks. STGCN ( Yu et al. (2018)) and DCRNN ( Li et al. (2018)) operate on static graphs and combine temporal convolution or recurrent modules with graph convolution for short-term forecasting. GMAN ( Zheng et al. (2020)), PDFormer ( Jiang et al. (2023)) are transformer-based methods model dynamic spatial dependencies via spatial self-attention. These models assume fixed connectivity between nodes, making them inapplicable to domains where the underlying network structure evolves. While some extensions like AGCRN ( Bai et al. (2020)), DSTAGNN ( Lan et al. (2022)), MTGNN ( Wu et al. (2020)) and Graph WaveNet ( Wu et al. (2019)) incorporate latent or adaptive graphs, they often do not model fully dynamic edge sets or permit per-timestep graph changes. TGAT ( Xu et al. (2020)) is another transformer-based model that applies temporal attention and time encoding to perform node classi-

fication and link prediction on a single evolving graph. However, it is not applicable to our setting, which involves forecasting node attributes across multiple graph instances, each with its own time-varying adjacency sequence. STG-NCDE ( Choi et al. (2022)) models traffic with neural controlled differential equations (NCDE), combining separate continuous-time spatial/temporal NCDEs; it also assumes a shared topology per dataset and static prior relationships. Another conceptually related method is AGATE ( Yamasaki et al. (2023)), which is a holistic framework for next-step graph evolution that jointly models node/edge birth–death and node-attribute dynamics via an interdependent ('reuse') stage. In contrast, our objective is specialized node-attribute forecasting under a fixed node set; we neither supervise nor evaluate link or node-birth processes.

**Node Attribute Prediction in Dynamic Graphs.** Surprisingly few works explicitly address multivariate node attribute prediction in dynamic graphs with changing edge structure. DGCRN ( Li et al. (2023)) is one of the few methods that allow learning dynamic relationships between nodes, but it still does not accommodate using prior knowledge of a dynamic graph as input.

Our work differs from prior approaches in three key ways:

- We formulate and tackle **per-sample dynamic-graph forecasting**: multi-step, multi-dimensional node-attribute prediction where each training example provides its own evolving topology.

- We introduce a **graph-portable spatial encoder** that injects the provided $A_t$ as an additive *edge-bias* in attention at each time step, preserving permutation equivariance and incorporating edge dynamics.

- We combine the encoder with a **rollout-robust temporal decoder** with scheduled sampling and horizon-aware loss and demonstrate consistent gains on heterogeneous datasets, including cases where per-sample graphs materially improve accuracy.

## 3 PROBLEM FORMULATION

Let $\mathcal{G}_t = (\mathcal{V}, \mathcal{E}_t)$ denote a graph snapshot at time $t$, where $\mathcal{V}$ is a fixed set of $N$ nodes and $\mathcal{E}_t$ is the edge set at time $t$. Let $X_t \in \mathcal{R}^{N \times D}$ be the matrix of node features, and $A_t \in \mathcal{R}^{N \times N}$ be the adjacency matrix corresponding to $G_t$ at time $t$. Given a sequence $\{X_1, \ldots, X_L\}$ and $\{A_1, \ldots, A_L\}$, the goal is to predict future node features $\{X_{L+1}, \ldots, X_{L+H}\}$, conditioned on both node feature evolution and the dynamic graph structures. An illustration of the problem can be found in Figure 1.

## 4 METHODS

### 4.1 OVERVIEW

We want multi-step, multi-dimensional node-attribute forecasts on time-evolving graphs, and we often have a different graph per training sample (e.g., per subject, per day). That asks for a spatial module that respects graph structure but stays permutation-equivariant and portable to new graphs, a temporal module that's stable for long horizons, and a decoder that handles distribution shift as we roll out predictions. DynaSTy's blocks map cleanly onto these needs. See Figure 2 for a high-level modular diagram of the architecture.

### 4.2 INPUT REPRESENTATION

Each training sample consists of:

- A node attribute history tensor $X_{\text{hist}} \in \mathcal{R}^{N \times D \times L}$, where $N$ is the number of nodes, $D$ is the feature dimension of each node and $L$ is the input sequence length.

- A dynamic graph sequence $A_{\text{hist}} \in \mathcal{R}^{N \times N \times L}$, representing one adjacency matrix per time step.

The forecasting target is a trajectory $Y \in \mathcal{R}^{N \times D \times H}$ of node attributes over $H$ future time steps.

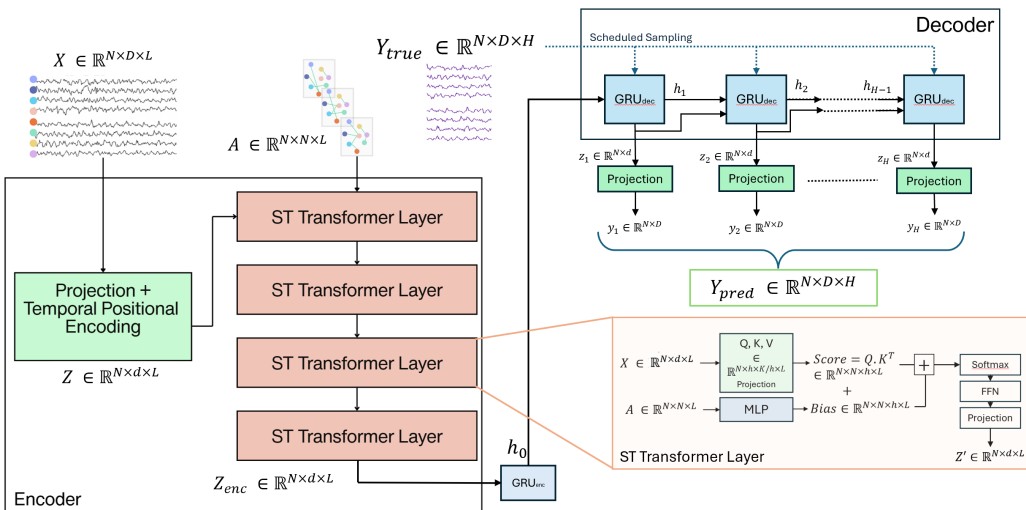

Figure 2: High-level Architecture Diagram

### 4.3 ENCODING AND TEMPORAL POSITION

We begin by linearly projecting each node's input feature vector to a hidden dimension $d$, and add learnable temporal positional encodings:

$$Z = \text{Linear}(X_{\text{hist}}) + \text{PE}_{\text{time}}$$

This produces $Z \in \mathcal{R}^{N \times d \times L}$, which is then passed through a stack of transformer layers. Even though the temporal model is a GRU/temporal block later, giving each step a distinctive "timestamp vector" helps the spatial encoder condition on phase (e.g., rush hour vs midnight) so it can form time-aware spatial contexts. Learned encodings can also absorb dataset-specific periodicities beyond simple sinusoids.

### 4.4 TRANSFORMER LAYERS

Each transformer layer integrates dynamic edge-aware multi-head attention by injecting per-time-step adjacency information as a learnable bias in the attention scores. This allows the model to adapt spatial attention to changing graph structure, which is critical for domains like brain networks and evolving trust graphs. At each layer, node $i$ attends to node $j$ at time $t$ using:

$$\text{score}_{ij} = \frac{Q_i^\top K_j}{\sqrt{d}} + \text{Bias}_{ij}(A_t)$$

where $\text{Bias}_{ij}$ is computed via a small MLP (2 layers, 64 dimensions each) over $A_t[i, j]$. The output is passed through a residual feedforward block. We apply edge dropout during training to encourage robustness to noise. The bias term acts as a *learned log-prior* over plausible message routes given the current graph, nudging attention toward neighbors that the topology deems influential, while the content term $\langle Q_i, K_j \rangle$ allows attention to override the graph when feature similarity indicates otherwise. This decouples *where to look* (softmax over biased logits) from *what to aggregate* (values), improving expressivity over fixed graph filters. Different heads can specialize to different regimes of $A_t$ (e.g., strong vs. weak ties), which a single global head cannot capture. Because the bias only depends on the permuted entries of $A_t$, the layer remains permutation-equivariant and naturally supports *per-sample* graphs by simply supplying each sample's $A_t$ sequence.

In the current architecture, the encoder only uses spatial attention, with temporal modeling delegated to the decoder. We also experimented with adding temporal self-attention in the encoder, which improved the performance, but minimally. These results are reported in the appendix.

## 4.5 FORECASTING DECODER

The output from the encoder, $Z_{\text{enc}} \in \mathcal{R}^{N \times d \times L}$, represents the encoded history of node features across the $L$ input time steps, where $N$ is the number of nodes and $d$ is the hidden dimension. This tensor is first reshaped and permuted into a format suitable for sequence modeling, and passed through a GRU encoder to obtain an initial hidden state $h_0$ that summarizes the historical dynamics of each node.

The decoder then operates autoregressively over the prediction horizon $H$. At each future time step $t$, the decoder GRU generates an output $z_t$ and new hidden state $h_t$ based on the previous hidden state and the current input. The output is passed through a forecast-step MLP to produce the predicted node features at time $t$, denoted as $y_t \in \mathcal{R}^{N \times D}$, where $D$ is the feature dimension. These outputs are collected over the entire forecast horizon to yield the final output tensor $Y_{\text{pred}} \in \mathcal{R}^{N \times D \times H}$.

**Scheduled Sampling.** During training, we employ *scheduled sampling* to bridge the gap between the training and inference conditions, a technique introduced by (Bengio et al. (2015)). In traditional teacher forcing, the decoder is always fed the ground truth from the previous time step. However, at inference time, ground truth is not available, and the decoder must rely entirely on its own predictions. To mitigate this train-test discrepancy, we probabilistically choose between using 1. only the decoder's previous prediction $\hat{y}_{t-1}$ and 2. a weighted combination of the ground truth $Y_{t-1}^{\text{true}}$ and the model's own previous prediction $\hat{y}_{t-1}$ as input at each time step during training. This probability is governed by a decaying function of the training epoch, such that early in training, the decoder relies mostly on ground truth, and gradually transitions to using its own predictions as training progresses. This improves robustness and reduces error accumulation over long forecasting horizons.

The complete predicted sequence is generated by unrolling the decoder in $H$ time steps, using sampled or predicted inputs, and projecting the hidden states back into the feature space through the MLP head.

## 4.6 LOSS FUNCTIONS

Our loss function is a combination of Mean Absolute Error and a Variation loss:

$$\mathcal{L} = \mathcal{L}_{\text{MAE}} + \lambda \cdot \mathcal{L}_{\text{var}}$$

where $\lambda$ is a weighting coefficient, and:

$$\mathcal{L}_{\text{MAE}} = \sum_{t=1}^{H} w_t \cdot \text{MAE}(Y_t^{\text{pred}}, Y_t^{\text{true}}) \tag{1}$$

$$\mathcal{L}_{\text{var}} = \sum_{t=1}^{H-1} \text{MAE}(Y_{t+1}^{\text{pred}} - Y_t^{\text{pred}}, Y_{t+1}^{\text{true}} - Y_t^{\text{true}}) \tag{2}$$

The weights $w_t$ are exponentially decaying to emphasize short-term accuracy. Variation loss penalizes differences in temporal derivatives (i.e., frame-to-frame changes) between the prediction and ground truth, and discourages oversmooth predictions by explicitly penalizing when the predicted signal lacks the expected variability over time.

## 4.7 MASKED PRETRAINING

To enhance the model's representation of spatiotemporal dependencies before supervised forecasting, we introduce a self-supervised masked pretraining objective. Inspired by masked language modeling in NLP (Devlin et al. (2019)), we randomly mask a subset of entries across nodes and time steps in the input history tensor $X_{\text{hist}} \in \mathcal{R}^{N \times D \times L}$ and train the model to reconstruct these values using the corresponding adjacency sequence $A_{\text{hist}}$. This technique improves representation learning and has shown success in both sequence modeling and graph neural networks ( Devlin et al. (2019); Liu et al. (2023); Hu et al. (2020)).

**Masking Strategy.** For each training sample, we generate a binary mask $M \in \{0, 1\}^{N \times D \times L}$ by sampling entries uniformly at random with probability $p_{\text{mask}} = 0.15$. The masked input $\tilde{X}_{\text{hist}}$ is

Table 1: Network Statistics

| Dataset | Nodes | Input length | Output length | Feature dimension |
|---------|-------|--------------|---------------|-------------------|
| **Bitcoin Alpha** | 1296 | 12 | 8 | 2 |
| **Bitcoin OTC** | 1304 | 12 | 8 | 2 |
| **METR-LA** | 207 | 12 | 6/12 | 1 |
| **PEMS-Bay** | 325 | 12 | 6/12 | 1 |
| **Brain** | 200 | 12 | 8 | 1 |

created by zeroing out the selected entries:

$$\tilde{X}_{\text{hist}} = X_{\text{hist}} \odot (1 - M)$$

We feed $\tilde{X}_{\text{hist}}$ and $A_{\text{hist}}$ into the encoder and decode a full reconstruction $\hat{X}_{\text{hist}}$ using the same projection head used in forecasting.

**Loss Function.** We compute a masked reconstruction loss that penalizes reconstruction error only at masked positions:

$$\mathcal{L}_{\text{pretrain}} = \frac{\|(\hat{X}_{\text{hist}} - X_{\text{hist}}) \odot M\|^2}{\|M\|_1 + \epsilon}$$

where $\epsilon$ is a small constant to avoid division by zero. This objective trains the model to learn generalizable spatiotemporal representations, even in the absence of forecasting supervision.

**Pretraining Schedule.** We first train the model for a fixed number of epochs using $\mathcal{L}_{\text{pretrain}}$ only, and then fine-tune on the forecasting task with the supervised loss $\mathcal{L} = \mathcal{L}_{\text{MAE}} + \lambda \mathcal{L}_{\text{var}}$. We observe that masked pretraining improves performance, particularly on datasets with noisy or irregular structure such as traffic and trust networks.

### 4.8 TRAINING PROCEDURE

We train using Adam with a learning rate of $10^{-3}$ for up to 100 epochs, using early stopping based on validation RMSE. During early epochs, we apply curriculum learning by gradually increasing the forecast horizon.

## 5 EXPERIMENTS

We run our models on different relevant datasets described below, and conduct analyses on effects of certain hyperparameters on training time and RMSE.

### 5.1 DATASET AND SETUP

**Bitcoin Trust Networks**: We evaluate our model on the dynamic Bitcoin-OTC and Bitcoin-Alpha trust networks, obtained from the Stanford Large Network Dataset Collection (Kumar et al. (2016; 2018)), which capture time-evolving ratings exchanged between users in peer-to-peer marketplaces. Each dataset consists of a temporal edge list where nodes represent users and edges encode ratings (ranging from -10 to 10) given by one user to another at a particular timestamp. We discretize time into fixed-size intervals and construct a dynamic graph sequence by aggregating ratings within each interval. At every time step, each user (node) is represented by a 2-dimensional feature vector: their average rating given and average rating received within that interval.

**Traffic data**: The METR-LA traffic dataset contains speed measurements from sensors distributed across the Los Angeles metropolitan area. The underlying network is static, defined by the physical distances between sensors (Jagadish et al. (2014)). To simulate dynamic topology, we generate semi-synthetic dynamic graphs by repeating the static network across all time steps. We use sequences of 12 time steps as input and predict the following 12 and 6 time steps. The PEMS-BAY dataset is a similar one, consisting of highway traffic sensor readings collected by the California Department of Transportation, processed and released by Li et al. (2018)

**Brain Networks**: We also evaluate our model on fMRI data from the ABIDE dataset (Di Martino et al. (2014)), where each subject has a time series of BOLD signals across a consistent set of brain regions. For each subject labeled as neurotypical (label = 0), we extract a matrix of shape [N, T], where N = 200 is the number of anatomical regions of interest (ROIs), and T is the number of fMRI time points. To construct dynamic brain networks, we apply a sliding window (size 20, stride 1) over each subject's time series, compute Pearson correlation matrices within each window, and threshold the absolute values at 0.8 to obtain binarized adjacency matrices. This yields a sequence of dynamic graphs [T', N, N] and corresponding node features [T', N, 1] per subject.

We split the datasets into training (70%), validation (10%), and test (20%) sets and normalize inputs. The output from the models are denormalized before calculating evaluation metrics.

Table 2: Performance (MAE and RMSE) of different methods on various datasets.

| Dataset | Bitcoin Alpha | | Bitcoin OTC | | METR (12 steps) | | METR (6 steps) | | PEMS (12 steps) | | PEMS (6 steps) | | Brain | |
|---|---|---|---|---|---|---|---|---|---|---|---|---|---|---|
| Method | MAE | RMSE | MAE | RMSE | MAE | RMSE | MAE | RMSE | MAE | RMSE | MAE | RMSE | MAE | RMSE |
| LSTM | 2.1 | 3.8 | 2.89 | 3.6 | 4.57 | 9.5 | 4.3 | 7.93 | 3.43 | 6.59 | 2.49 | 4.68 | 26.66 | 37.1 |
| STGCN | 2.8 | 3.65 | 2.09 | 4.7 | 4.5 | 9.5 | 3.49 | 7.34 | 2.52 | 5.66 | 1.81 | 4.36 | 25.53 | 36.8 |
| DCRNN | 1.18 | 3.42 | 1.43 | 2.8 | 3.6 | 7.6 | 3.15 | 6.45 | 2.07 | 4.74 | 1.74 | 3.97 | 23.47 | 36.5 |
| Graph WaveNet | 1.27 | 2.85 | 1.38 | 2.86 | 3.53 | 7.37 | 3.07 | 6.22 | 1.95 | 4.52 | 1.63 | 3.70 | 22.54 | 33.75 |
| MTGNN | 2.17 | 3.19 | 2.33 | 3.21 | 3.49 | 7.23 | 3.05 | 6.17 | 1.94 | 4.49 | 1.65 | 3.74 | 21.9 | 35.1 |
| DGCRN | 1.54 | 2.9 | 1.56 | 2.53 | 3.44 | 7.1 | 2.99 | 6.05 | 1.89 | 4.42 | 1.59 | 3.63 | 26.19 | 39.35 |
| PDFormer | 1.53 | 3.09 | 1.67 | 2.98 | 3.31 | 6.98 | 2.83 | 5.96 | 1.89 | 3.8 | 1.61 | 3.3 | 25.7 | 36.5 |
| staticDynaSTy | 1.57 | 3.01 | 1.41 | 2.81 | **3.23** | **6.35** | **2.48** | **5.4** | **1.85** | **3.73** | **1.48** | **2.96** | 20.51 | 27.4 |
| shuffledDynaSTy | 1.36 | 3.1 | 1.38 | 2.51 | **3.23** | **6.35** | **2.48** | **5.4** | **1.85** | **3.73** | **1.48** | **2.96** | 20.5 | 28.32 |
| **DynaSTy** | **1.15** | **2.8** | **1.37** | **2.49** | **3.23** | **6.34** | **2.48** | **5.4** | **1.85** | **3.73** | **1.48** | **2.96** | **17.61** | **26.84** |

## 5.2 BASELINES AND METRICS

We compare our model against the following widely used spatiotemporal baselines:

- **LSTM** Hochreiter & Schmidhuber (1997): A fully connected long short-term memory network that ignores graph structure and models the temporal dynamics of each node independently. It serves as a strong non-graph baseline for time series forecasting.

- **STGCN** Yu et al. (2018): Combines spectral graph convolution with temporal gated convolutions. Assumes a fixed graph structure and models spatial and temporal dependencies separately using CNN-based operations.

- **DCRNN** Li et al. (2018): Introduces diffusion convolution over a static graph into a recurrent neural network, enabling spatiotemporal sequence modeling through localized message passing and gated recurrence.

- **Graph WaveNet** Wu et al. (2019): Uses adaptive graph learning and wavelet-inspired temporal convolutions to model spatiotemporal dynamics. Like other baselines, it assumes a global graph, either fixed or learned.

- **MTGNN** Wu et al. (2020): Learns a static graph structure and temporal dependencies jointly using graph attention and dilated temporal convolutions. While it can learn the graph, it still assumes a single global graph shared across samples. MTGNN offers users the option to provide a predefined graph as well, and all of our reported results are the best of the two versions.

- **DGCRN** (Li et al., 2023). A recurrent encoder–decoder that *generates a dynamic adjacency at each time step* via a hyper-network (conditioned on features/hidden states) and fuses it with a pre-defined static graph.

- **PDFormer** (Li et al., 2023). Models spatiotemporal data by combining a transformer encoder with learned spatial priors, where these priors modulate the attention weights so that the model can focus on spatially meaningful relationships while capturing temporal dynamics through self-attention. Unlike DynaSTy, it does not support per-sample or explicit time-varying adjacency matrices as input, but represents a strong static-graph attention-based baseline.

We evaluate all models using Mean Absolute Error (MAE) and Root Mean Squared Error (RMSE). Classical univariate forecasting methods such as ARIMA and SARIMA are not included, as they are not designed for high-dimensional, interconnected systems and have been shown to perform poorly

(Li et al. (2018); Wu et al. (2019)) in high-dimensional, non-stationary graph settings like those we study.

The mechanism of converting dynamic graphs to static for running baseline methods that operate on static graphs is described in Appendix A.

## 6  RESULTS

Mean Absolute Error (MAE) and Root Mean Squared Error (RMSE) results are presented in Table 2. The model used had 4 attention heads, 4 transformer layers, 48 hidden dimensions for all the datasets. All reported values are means over 10 independent runs, with standard deviations between 0.0009 and 0.06. Our method significantly outperforms existing approaches on all datasets, with p-values between 0.000007 and 0.00012.

To demonstrate the effectiveness of considering dynamic graphs instead of static, we ran our model on the aggregated static graphs that were used in the other baseline models and report these results next to the *staticDynaSTy* method in Table2. We observe that the dynamic version outperforms the static when the original graphs are in fact dynamic, i.e., the Bitcoin and brain networks.

**Per-sample graphs matter.**  To test whether conditioning on each sample's own dynamic graph helps, we shuffled the input dynamic graphs at training time so that the node dynamics and the dynamic graphs do not correspond anymore. These results are in the second last row of Table 2. We can see that in Bitcoin and Brain datasets, the performance significantly drops as a result of this shuffling, indicating that we are losing information. In case of the traffic datasets, no such drop is observed because these graphs by design are shared globally and no per sample graphs are available.

As a second test, we compare DynaSTy with the next-best performing method, Graph WaveNet, using a brain fMRI dataset with two cohorts: neurotypical ($y{=}0$) and neurodivergent ($y{=}1$). **DynaSTy**, which consumes each subject's dynamic graph sequence $A_{1:H}$, attains RMSE=26 when trained/evaluated only on $y{=}0$, RMSE=21 only on $y{=}1$, and 22 when trained jointly on the combined cohort. Joint training therefore yields a $15.4\%$ reduction relative to $y{=}0$ alone ($26 \rightarrow 22$) while remaining within $4.8\%$ of the $y{=}1$-only optimum ($21 \rightarrow 22$). In contrast, **Graph WaveNet**, which learns a single shared adjacency for all samples, yields RMSE=36 on $y{=}0$, 29 on $y{=}1$, and 33 on the combined cohort. Joint training thus helps the harder cohort ($36 \rightarrow 33$) but does not approach the easier cohort's optimum (29), indicating that a single global graph forces a compromise that cannot capture cohort-specific connectivity.

### 6.1  HYPERPARAMETER SENSITIVITY

The main hyperparameters in DynaSTy are the number of attention layers, number of attention heads in each layer and the number of hidden dimensions the input is projected to. We ran our model on the METR-LA and brain datasets with varying each hyperparameter while keeping the others constant and documented the effect on training time and RMSE as shown in Figure 3. It is observed that while all three parameters have a significant effect on RMSE, increasing the hidden dimension seems to be the most effective, while also maintaining the training time within a reasonable range (maximum 102 seconds for METR-LA and 73 seconds for Brain). On the other hand, increasing the number of attention heads or the number of transformer layers, both related to the transformer component, quickly increases training time to about 125 seconds for METR-LA and 140 seconds for Brain.

We also document the average training times per epoch and inference times for DynaSTy and compare with baselines on the METR-LA dataset with an input sequence length of 12 and output (prediction) sequence length of 12 in 4 in Appendix B.

## 7  CONCLUSION AND FUTURE WORK

We introduce DynaSTy, a spatiotemporal transformer architecture for node attribute prediction on dynamic graphs. The model combines a dynamic edge-bias attention mechanism with an autoregressive GRU-based decoder to jointly capture spatial and temporal dependencies. Empirical results

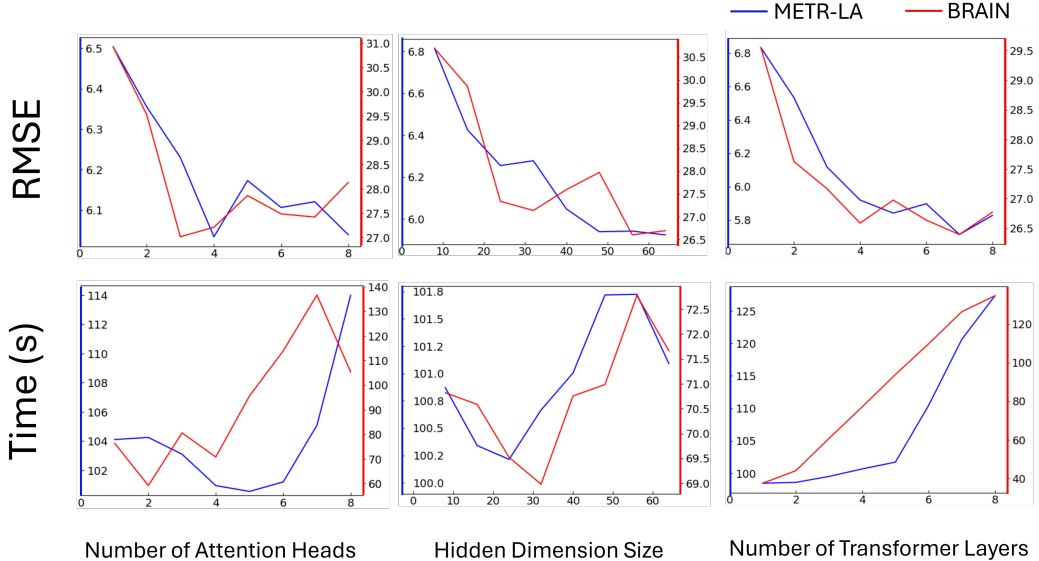

Figure 3: Impact of architectural hyperparameters on performance and runtime.

This figure shows the effect of the number of attention heads (left), hidden dimension size (center), and number of transformer layers (right) on both RMSE and training time for the METR-LA and BRAIN datasets with input sequence of length 12, and outputs of length 12 and 8 respectively. For each subplot, the primary Y-axis (blue) corresponds to the METR-LA dataset, and the secondary Y-axis (red) corresponds to the BRAIN dataset.

Table 3: Ablation study with edge bias, masked pretraining and variation loss. Table shows mean RMSE values across 10 runs.

| Configuration | Brain (8 steps) | Brain (12 steps) | METR (6 steps) | METR (12 steps) |
|---|---|---|---|---|
| **Full Model** | 26.84 | 27.29 | 5.4 | 6.34 |
| **w/o Edge bias** | 31.92 | 33.26 | 6.21 | 7.53 |
| **w/o Pretraining** | 28.53 | 29.52 | 5.88 | 7.01 |
| **w/o Variation Loss** | 27.02 | 30.31 | 5.79 | 7.25 |

on real-world network datasets with evolving edge structures demonstrate consistent improvements over strong spatiotemporal baselines, including DCRNN and MTGNN.

In future work, we aim to extend this framework in several directions. First, we plan to apply DynaSTy to more complex real-world dynamic systems where the temporal resolution may vary across time or across entities. Second, we intend to develop a distributed version of DynaSTy to improve scalability on large-scale networks, such as Bitcoin trust graphs with over 1,000 nodes, where current runtimes are a bottleneck. Another way of reducing the runtime complexity would be by shifting to a sparse attention paradigm instead of full attention which is $O(N^2)$.

Finally, a key limitation of the current design is its assumption that the node set remains fixed over time. This presents two challenges: (i) it implicitly assumes that all nodes exist from $t = 0$, which is often unrealistic in dynamic systems such as social or biological networks where nodes appear and disappear; and (ii) it imposes computational and memory overhead by forcing inactive or disconnected nodes to be included at every time step. Extending DynaSTy to support a variable node set over time would require a transition from discrete-time snapshot modeling to a continuous-time representation of dynamic graphs, which we consider a promising and important future research direction.

## 8 REPRODUCIBILITY STATEMENT

All code and link to datasets used are available in https://anonymous.4open.science/r/DYNASTY-4DFD/ and the instructions to run them will be updated soon.

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

Our experiments were implemented in Python 3.10 using the PyTorch deep learning framework. All experiments were run on a Linux server with CUDA 11.8 and NVIDIA H100 GPUs.

## A    STATIC GRAPH ASSUMPTIONS IN BASELINES.

All the STGNN baselines considered (DCRNN, STGCN, MTGNN) require a single static adjacency matrix that is shared across time and across training examples. In contrast, our model is designed to handle dynamic graphs that vary per time step and per sample. To ensure a fair comparison, we construct fixed adjacency matrices tailored to each dataset. MTGNN is slightly better because it offers the option to learn the adjacency matrix from the observed data, but it is still static and shared across samples.

**Brain Networks.** In the brain network setting, each subject yields a distinct dynamic graph sequence derived from pairwise correlations between brain regions. To aggregate these into a single global adjacency matrix suitable for the baselines, we compute a thresholded binary correlation matrix for each subject and then average these across subjects to produce an edge co-occurrence matrix. We then threshold this matrix at $\tau = 0.5$, retaining edges that appear in more than 50% of individual graphs. This consensus adjacency matrix captures the most consistent inter-regional connections across the population and is used as the fixed graph for all static-graph baselines.

**Bitcoin Trust Networks.** In contrast, the Bitcoin trust networks involve user-to-user interactions recorded over time. Here, we construct the fixed graph by including an edge between two users if they ever interacted (i.e., if any trust rating was exchanged between them at any time). This results in a binary adjacency matrix capturing the union of all observed edges across the dataset. The loss of information is less concerning in this case since the training samples belong to a single dynamic system, unlike the brain dataset where each training sample came from a different person.

These distinct aggregation strategies reflect the differing structure and semantics of the two domains: the brain networks involve latent, population-level structure shared across subjects, while the Bitcoin networks reflect explicit interaction histories between individual agents.

## B    RUNTIME COMPLEXITY

We analyze the computational complexity of DynaSTy in terms of the key input parameters: batch size $B$, number of nodes $N$, input sequence length $L$, hidden dimension $d$, number of attention heads $h$, and prediction horizon $H$.

**Input Projection and Positional Encoding.** The linear projection of node attributes from input dimension $D$ to hidden dimension $d$ and the addition of learnable positional encodings incurs a cost of:

$$\mathcal{O}(B \cdot L \cdot N \cdot d)$$

**Spatiotemporal Transformer Layers.**  Each transformer layer applies edge-biased multi-head attention across all node pairs per time step. This involves:

- Attention score computation: $\mathcal{O}(N^2 \cdot d)$ per time step
- Edge-bias MLP over each pairwise edge: $\mathcal{O}(N^2 \cdot h)$

Summed across the batch and sequence length, the total complexity is:

$$\mathcal{O}(B \cdot L \cdot N^2 \cdot (d + h))$$

**GRU Encoder.**  We flatten the sequence across time and encode each node's $L$-step history using a GRU, with complexity:

$$\mathcal{O}(B \cdot N \cdot L \cdot d^2)$$

**GRU Decoder.**  The decoder autoregressively predicts $H$ future steps per node, each using GRU recurrence and MLP projection:

$$\mathcal{O}(B \cdot N \cdot H \cdot d^2)$$

**Output Projection.**  Mapping decoder outputs back to the original feature space costs:

$$\mathcal{O}(B \cdot H \cdot N \cdot d)$$

**Overall Complexity.**  The total runtime is dominated by the spatiotemporal attention and decoder GRU steps:

$$\mathcal{O}\big(B \cdot L \cdot N^2 \cdot d\big) + \mathcal{O}\big(B \cdot N \cdot (L + H) \cdot d^2\big)$$

**Scalability Consideration.**  The quadratic dependency on the number of nodes $N$ in the attention mechanism may limit scalability on large graphs (e.g., Bitcoin trust networks with $N > 1000$). Future work may explore sparse or localized attention to improve computational efficiency.

### B.1    COMPARISON OF RUNTIMES

Since DynaSTy has a pretraining step with 15 epochs, this time was added to the total training time before averaging across epochs. Table 4 reports this pretraining + training time in seconds. The main bottleneck in the training time are the transformer layers, which makes the model scale quadratically in the number of nodes. It also scales quadratically in the number of hidden dimensions, but most often the number of nodes is much greater than the hidden dimension, especially in cases like the Bitcoin networks.

Table 4: Wall Times on the METR-LA dataset averaged across 50 epochs

| Method | Training (s) | Inference (s) |
|---|---|---|
| STGCN | 54 | 12 |
| DCRNN | 320 | 92 |
| Graph WaveNet | 187 | 52 |
| MTGNN | 173 | 48 |
| DGCRN | 155 | 42 |
| PDFormer | 132 | 46 |
| DynaSTy | 124 | 33 |

## C    TEMPORAL SELF ATTENTION

Given encoder activations $H \in \mathbb{R}^{B \times N \times d \times L}$ (batch $B$, history length $L$, nodes $N$, channel dimension $d$), we optionally add a lightweight *temporal* block that attends *within each node* across time, without mixing nodes. This module models long-range and non-uniform temporal dependencies (variable lags, periodicities) at each node, complementing the spatial encoder that conditions on $A_t$ per step. Computationally, it costs $\mathcal{O}(BNL^2D)$ (attention over $L$ for each of $BN$ node streams)

Table 5: Effect of temporal self-attention on RMSE ($\downarrow$). Same encoder/decoder, training schedule, and data splits; only the temporal-attention block is toggled.

| Dataset | RMSE w/o Temp. Attn. | RMSE w Temp. Attn. | $\Delta$ (%) |
|---|---|---|---|
| Bitcoin Alpha | 2.8 | 2.6 | -7.14 |
| Bitcoin OTC | 2.49 | 2.38 | -4.41 |
| METR-LA | 6.34 | 6.1 | -3.78 |
| PEMS-Bay | 3.73 | 3.71 | -0.53 |
| Brain | 26.84 | 26.11 | -2.72 |

and preserves permutation equivariance over nodes. In our implementation it can be toggled on/off; when enabled, we insert it after the spatial transformer stack and before the GRU summarizer, keeping the rest of the architecture unchanged. We observed a 0.53% - 7.1% reduction in RMSE on our datasets, reported in Table 5.

