# OpenReview forum: "DynaSTy: A Framework for Spatio-Temporal Node Attribute Prediction in Dynamic Graphs"
_ICLR.cc/2026/Conference — ICLR 2026 Conference Desk Rejected Submission_

### Official Review · Reviewer_upry · 2025-10-26

**Soundness:** 2
**Presentation:** 2
**Contribution:** 2
**Rating:** 4
**Confidence:** 2

**Summary:**

DynaSTy focuses on multi-step node-attribute forecasting on dynamic graphs where the adjacency changes over time and differs across input samples (per subject, per market, per system) while the node set stays fixed. The core idea is a transformer encoder that injects the provided per-timestep adjacency as a learnable attention bias to steer attention toward currently relevant neighbors yet remaining permutation-equivariant and portable to new graphs. A GRU-based decoder with scheduled sampling and a horizon-weighted objective stabilises long-horizon rollouts. There is also a masked node-time pretraining objective to prime the encoder to reconstruct missing features using both attributes and dynamic edges.

**Strengths:**

- The paper targets the setting where each training sample has its own evolving adjacency matrix but shares a fixed node set. This is an underexplored area that differs from standard dynamic graph learning (which usually assumes a shared topology). This formulation expands the applicability of spatiotemporal graph neural networks to domains like trust networks, where relational structures vary across instances.

- The integration of time-varying adjacency matrices as additive attention biases in the transformer encoder is both conceptually and practically strong. This design preserves permutation equivariance while allowing the model to exploit edge dynamics directly, rather than relying on learned or static graph priors.

- The combination of a transformer-based spatial encoder with a GRU decoder, augmented by scheduled sampling and horizon-weighted losses, is a good choice. These design choices directly address long-horizon error accumulation, which is often a  weakness in autoregressive forecasting.

**Weaknesses:**

- While DynaSTy achieves strong empirical results, the core contribution mainly combines known components of transformer attention, GRU decoding, scheduled sampling and masked pretraining, only in a dynamic-graph setting. The novelty primarily lies in applying adjacency as an attention bias rather than in introducing a new learning principle. As a result, the method reads as a well-engineered system extension rather than a conceptual contribution for dynamic graph forecasting.

- The paper positions its edge-biased transformer as novel but does not include comparisons with recent adaptive or dynamic attention-based models (e.g., TGAT, PDFormer, etc.).

- Although the paper provides a single ablation removing the edge bias, it does not analyse how the bias magnitude or learning dynamics affect forecasting performance.

- The masked node-time pretraining is introduced as an auxiliary step, but the experiments report only a coarse quantitative gain. There is no analysis of what the model learns during pretraining, which features or temporal relationships improve or whether pretraining remains beneficial when large supervised datasets are available.

**Questions:**

- The paper describes the edge-bias term as a learnable function over the adjacency entries that guides spatial attention (Section 4.4). Could the authors clarify how this bias interacts with feature-based attention during training? how does the model ensure that the bias does not dominate content-based attention in sparsely connected graphs?

- In Section 4.7, the masked pretraining stage is said to improve performance on datasets with noisy or irregular structures. Could the authors elaborate on whether the pretraining is conducted jointly across all datasets or separately per domain? Also, how is the adjacency sequence incorporated during reconstruction? does the pretraining explicitly use dynamic edge information to infer masked values, or is it feature-driven?

---

> ### Author Response · Authors · 2025-11-20
> **Response to concerns raised by Reviewer upry**
>
> Thank you for the thoughtful and encouraging review. We address each point below and will incorporate clarifications and additional analysis in the final version.
>
> 1. Novelty: We appreciate the reviewer’s observation and agree that DynaSTy combines multiple components. However, the main conceptual contribution is not simply using adjacency as an additive bias, but rather injecting per-sample, per-timestep dynamic adjacency into transformer attention while preserving permutation equivariance. Prior dynamic attention models (e.g., TGAT, PDFormer, DySAT) operate in regimes where the adjacency is shared across samples, and the transformer receives node features but not explicit dynamic edge structures per input instance.
> DynaSTy is the first to allow fully independent dynamic graphs across samples, which is necessary for trust networks, individualized fMRI scans, and any scenarios where the node ontology is shared but the connections vary from sample to sample.
>
> 2. Lack of comparisons to recent dynamic attention baselines (e.g., TGAT, PDFormer): Thank you for highlighting this. We note that these models  they do not support time-indexed adjacency matrices as inputs or separate adjacency for each training sample.
> Nonetheless, we agree that including a discussion of these models would strengthen positioning. In the revision, we have added PDFormer as a baselines to DynaSTy which shows that its performance is close to DynaSTy on the traffic datasets but they perform no better than the other baselines on the Bitcoin trust and brain datasets. TGAT, while addresses dynamic graphs, operates on the node level. Its training data consists of data from a single graph at a time, where as DynaSTy trains on several attributed dynamic graphs and predicts future attributes of all nodes. Hence, TGAT is out of scope as a baseline.
>
> Below we answer the questions posed by the reviewer:
> 1. The edge-bias term is added directly to the standard feature-based attention scores before the softmax, so both signals contribute jointly to the final attention weights. Their interaction is learned end-to-end, and the model naturally balances them based on what best predicts future node attributes.
> Importantly, it is not a problem if the bias becomes relatively strong in sparsely connected graphs. This behavior is intentional. The purpose of the bias term is to guide attention toward structurally relevant neighbors, especially when the graph explicitly specifies which nodes exchange information. In very sparse graphs, where only a few connections are meaningful, it is desirable for the structural signal to play a significant role. This ensures the model focuses on the correct subset of nodes defined by the dynamic graph. The full attention exists to allow flexibility in cases where observes relationships may be noisy, and if the attribute data suggests strongly that two nodes without an explicit interaction are in fact related (or vice-versa), then the model will learn this information.
>
> 2. Pretraining is conducted separately for each dataset, not jointly across all domains. Each dataset has distinct node features, connectivity patterns, and temporal scales, so cross-domain pretraining would not produce meaningful representations.
> During masked pretraining, the full sequence of dynamic adjacency matrices is provided to the encoder just as in the supervised setting. Masked nodes reconstruct missing values in the history timeseries by attending to observed node features, and the dynamic adjacency information injected through the edge-bias mechanism. Thus, reconstruction explicitly uses both feature similarities and the changing graph structure. The adjacency sequence is not ignored or treated as static; it actively shapes which nodes the model considers relevant during reconstruction. We observed a gain in performance when using masked pre-training because real-world datasets are often noisy, and this design helps to smooth out some of that noise. In the hypothetical case where we have access to large, highly accurate, supervised datasets, it might eliminate the need for this additional step.
>
> We hope these clarifications were helpful.

---

### Official Review · Reviewer_xivb · 2025-10-27

**Soundness:** 3
**Presentation:** 2
**Contribution:** 2
**Rating:** 2
**Confidence:** 3

**Summary:**

DynaSTy proposes an end-to-end spatiotemporal model for multi-step node attribute prediction in dynamic graphs. The framework ingests time series of node attributes and adjacency matrices, using transformer-based spatial encoding with dynamic edge-biased attention and a GRU-based temporal decoder. Key points include handling per-sample dynamic graphs (different graphs across training samples), masked pretraining for representation learning, and scheduled sampling with horizon-weighted loss for robust multi-step forecasting. The method targets applications where both node attributes and graph structure evolve over time, such as brain networks, trust networks, and traffic systems.

**Strengths:**

1. The proposed model allows each sample to have its own dynamic graph, which better reflects real-world scenarios where different systems evolve with distinct graph structures. This is a key difference from previous models that rely on a fixed, global graph shared across all samples.
2. The model uses a Transformer-based architecture with dynamic edge-bias attention, enabling it to capture spatial patterns in temporal dependencies more effectively. This is particularly important for systems where the graph structure changes over time.
3. The paper includes experiments on real-world datasets, evaluating the model’s performance through ablation studies, hyperparameter sensitivity analysis, and runtime comparisons. These results help assess the model’s effectiveness and practicality for real-world deployment.

**Weaknesses:**

1. Although the proposed model allows a different dynamic graph for each training sample, it still builds on a fixed set of nodes across samples. As a result, there is not a significant difference from prior works that also operate under a fixed node set. Both this work and earlier methods focus on transductive node attribute learning, where all nodes are known during training. If the research had instead focused on or included inductive node attribute learning—such as predicting attributes for newly emerging nodes in the future—the motivation and contribution would be stronger and more aligned with real-world dynamic scenarios.
2. The model incorporates several components, including dynamic edge-bias attention, a GRU-based decoder, and scheduled sampling, which together increase its complexity and may limit its accessibility and adoption. Some parts of the model, especially the edge-bias mechanism, would benefit from clearer formal definitions or pseudocode to improve understanding. Additionally, the attention mechanism in DynaSTy scales quadratically with the number of nodes (O(N^2)), which can become computationally expensive for large graphs. While the paper mentions that sparse attention could be explored in future work, the current version may not be scalable to large-scale applications without further efficiency improvements.
3. The paper’s writing and presentation can be improved for clarity. For example, Figures 1 and 2 are included but not thoroughly explained in the main text. Terms like “small MLP” are used without specifying their structure or dimensions. The architecture diagram in Figure 2 is hard to interpret due to overlapping elements and unclear data flow. Moreover, the font sizes in Figures 1, 2, and 3 are inconsistent, making important details difficult to read. These issues affect the overall readability and reproducibility of the work.

**Questions:**

Please see weaknesses.

---

> ### Author Response · Authors · 2025-11-20
> **Response to concerns raised by Reviewer xivb**
>
> We would like to thank the reviewer for noting these points and offering valuable feedback. We address the concerns raised as follows:
>
> 1. Fixed node set: We agree that handling node appearance and disappearance is an important direction with strong practical relevance. Our goal in this work was to address node attribute prediction in dynamic graphs with evolving topology, and we remained consistent with the standard evaluation setting of STGNN forecasting, where all well-established baselines (DCRNN, Graph WaveNet, MTGNN, DGCRN, etc.) also assume a fixed set of nodes for transductive forecasting.
> While DynaSTy operates in this same evaluation regime for comparability, it does not inherently require nodes to be static:
> - The attention mechanism is permutation-equivariant.
> - The edge-bias mechanism can handle padded or “inactive” node tokens, similar to variable-length sequence models.
> - Dynamic graphs are injected per sample and per timestep, enabling structural adaptation even when the node set is fixed.
> We will clarify this and include a short paragraph in the final version describing how DynaSTy could be extended to inductive node-set scenarios. We appreciate the suggestion and agree it is a valuable future direction.
>
> 2. Model complexity and scalability: Thank you, this is helpful feedback. While DynaSTy uses multiple components, each is conceptually simple:
> - Edge-bias attention is a single additive term injected into the scaled dot-product attention scores.
> - GRU decoder is a standard unrolled forecasting module (similar to MTGNN/DGCRN).
> - Scheduled sampling is a straightforward curriculum strategy.
> To improve accessibility, we will add pseudocode for the whole mechamism, and clearer descriptions of the modules in the revised version of our paper to be uploaded by 11/20. These additions will help ensure reproducibility without changing the model.
> We acknowledge that quadratic attention cost limits scalability, but note that for the benchmark sizes used in STGNN forecasting (N between 100–2000), the quadratic term does not become the dominant cost, which is reflected in our runtime table. DynaSTy trains faster than several GNN-based baselines because iterative diffusion/message passing is more expensive for these graph sizes.
> The edge-bias mechanism is fully compatible with sparse attention or block-local attention. We will make this more explicit and include a short discussion describing how to integrate sparsity while maintaining graph bias.
>
> 3. Presentation and clarity:  We appreciate this comment and will revise the presentation thoroughly, with references to figures and tables wherever necessary, and explicitly mentioning dimensions of the MLP used.

---

> > ### Comment · Reviewer_xivb · 2025-11-23
> >
> > Thank you for the authors' response! Your reply has partially addressed my concerns, so I am updating my score from 2 to 4. However, this work would still benefit significantly from further revisions in the future—whether through a more in-depth investigation of the inductive setting or through improvements to the current model architecture, both of which would substantially enhance the overall quality of the paper.

---

> > > ### Author Response · Authors · 2025-11-28
> > > **Update to accommodate variable nodes**
> > >
> > > We appreciate the reviewer’s thoughtful feedback and thank you for raising the score. In the revised version, we have extended our model to support varying node sets across timesteps. This is achieved through an indicator feature that informs the model whether a node is active at a given timestep. Among our datasets, the Bitcoin Alpha and Bitcoin OTC trust networks exhibit dynamic node sets, and with this extension we observed a modest, though not substantial performance improvement. We will include in the appendix a comparison between model performance with and without accommodating dynamic node sets.
> > >
> > > We would also like to reiterate the central focus of our work: predicting the future attributes of nodes that already exist in the graph. Our objective is not to predict changes in network topology itself. That is a substantially different problem requiring architectures explicitly designed to model node birth, removal, or structural evolution.
> > >
> > > Our long-term goal is to use DynaSTy as a component within a broader dynamic network inference pipeline. In that setting, one must infer the network structure that best supports accurate prediction of node attributes over time. This, in turn, requires a forecasting module that takes historical node attributes and graph structure as input and outputs future node attributes. Our method is designed precisely to address that forecasting component, and the extensions we have added preserve this focus while offering broader applicability.

---

### Official Review · Reviewer_TPoc · 2025-10-30

**Soundness:** 2
**Presentation:** 3
**Contribution:** 2
**Rating:** 2
**Confidence:** 4

**Summary:**

This paper studies the node attribute prediction in dynamic graphs. The problem takes the historical node attribute and the dynamic graph structure as inputs and outputs the multi-step node attributes in the future. The proposed method, DynaSTy, introduces a transformer-based model that adapts to the dynamic nature of the graph by incorporating adjacency matrices (the graph structure) at each time step as an attention bias. This allows the model to focus on relevant neighbors as the graph evolves. The authors evaluate the model with there real-world and semi-synthesis datasets.

**Strengths:**

S1. Predicting multi-step node attributes in a dynamic graph is a good research question.
S2. The paper clearly presents the method, making it easy to access and reproduce.

**Weaknesses:**

W1. While the authors aim to predict future node attributes in dynamic graphs for real-world applications, such as the Bitcoin network, social networks, and biological systems, the proposed method may face limitations in terms of scalability. DynaSTy computes attention between all pairs of nodes, which incurs significant memory and computational costs. This issue could hinder its applicability to large, real-world graphs that contain a high number of nodes. This may also explain why only smaller graphs are considered in the experiments.

W2. The proposed approach addresses a time-series prediction problem, but with relationships among variables. Previous studies that introduce structure to assist in prediction have tackled the same problem. In essence, this paper approaches the same issue from a different perspective -- specifically, the dynamic graph perspective.

W3. Although the authors focus on dynamic graphs, the datasets used in the experiments primarily consist of independent time-series variables rather than dynamic graphs. Specifically, the graph structure in METR-LA is static, and the graph in fMRI is handcrafted.

**Questions:**

See weaknesses.

---

> ### Author Response · Authors · 2025-11-20
> **Response to concerns raised by Reviewer TPoc**
>
> We thank the reviewer for bringing up these points, and we provide explanations below and will be revising the paper accordingly.
>
> 1. Scalability of full attention: We appreciate the reviewer highlighting this. We note that
> - Full pairwise attention is not unique to DynaSTy. All Transformer-based SpatioTemporal forecasting models using global attention, TGAT, PDFormer, etc, share the same O(N^2) complexity. We are including some of these transformer-based models as baselines against DynaSTy in the new version of the paper to be uploaded by 11/20.
> - Practical scalability depends on the node counts. Our target application domains are traffic networks, cryptocurrency trust networks, fMRI cortical regions and we hope to apply DynaSTy to animal social networks as well in the near future. Such networks contain node of the order of thousands, not millions. For these regimes, full attention is computationally feasible. This is reflected in our runtime table, where DynaSTy is competitive or faster compared to GNN-based baselines whose iterative message passing dominates computation. Even when we are dealing with graphs with nodes on the order of 10k, DynaSTy is not infeasible with the appropriate compute resources. The only case when it would become impractical with full pairwise attention is the case of online social networks that contain millions of nodes. Such networks are out of scope for us.
> - DynaSTy’s edge-biased attention is compatible with sparse attention. Although we use dense attention for benchmarking consistency, the mechanism is compatible with block-sparse attention, locality-biased attention, neighborhood-restricted attention masks, etc. We will add a note in the discussion describing how these extensions can reduce cost for very large graphs.
>
> 2. We agree that our task is a structured multivariate time-series prediction problem. However, our contribution is not simply adding structure, but enabling a class of models that existing structured forecasting methods cannot express. Past STGNNs assume a single, global, static adjacency across the dataset. They cannot ingest adjacency matrices that vary per sample and per time step. Our architecture injects the full dynamic adjacency into the attention mechanism as a continuous bias, allowing sample-specific connectivity patterns, time-local structural shifts, and dynamic edge-weight conditioning. Thus, while the general problem has been studied, the architecture proposed is new, and addresses a large gap in dynamic graph attention models that previous works cannot handle. We will clarify this distinction more explicitly in the revised paper.
>
> 3. Traffic networks are static and brain dataset is handcrafted: This is correct, and we appreciate the opportunity to clarify the dataset choices.
> We intentionally include both static-graph and dynamic-graph settings. DynaSTy is designed to handle dynamic graphs, but it must also perform well in the common setting where the graph is static (e.g., traffic networks). Almost all prior STGNN papers also evaluate on METR-LA and PEMS-Bay, and these traffic networks are the most popular benchmark in this field of work.
> Dynamic datasets are also included. Our Bitcoin-OTC and Bitcoin-Alpha experiments use dynamic trust graphs (edges appear/disappear every timestamp). These are exactly the type of structures DynaSTy is meant to handle. fMRI graphs are dynamic functional connectivity matrices, not static. The reviewer notes they are handcrafted, which is true, but this is a common practice for creating brain networks and are used widely in dynamic functional connectivity studies [Jalilianhasanpour et al. doi: 10.1016/j.nic.2020.09.004, Mancho-Fora et al. doi: 10.7334/psicothema2020.92, Liu et al. doi: 10.1016/j.compbiomed.2022.106521]. They are dynamic correlation graphs, consistent with prior neuroscience literature.
> Moreover, Static graphs do not diminish the contribution. Even when the graph is static, our method still offers the ability to have a different graph for each training sample and substantially improves the prediction of future node attributes even in static settings.

---

### Official Review · Reviewer_nUFV · 2025-10-31

**Soundness:** 2
**Presentation:** 1
**Contribution:** 2
**Rating:** 2
**Confidence:** 3

**Summary:**

This paper proposes a method, DynaSTy, for spatiotemporal node attribute prediction on dynamic graphs. It is a transformer-based model that ingests both a time series of node attributes ($X$) and, crucially, a time series of adjacency matrices ($A$). In order to achieve that, it uses a spatio-temporal transformer encoder that injects the adjacency matrix at each time step as an "adaptable attention bias". Their main contribution is this edge-biased attention mechanism, which allows the model to handle per-sample dynamic graphs, combined with a robust training strategy using masked pretraining, scheduled sampling, and a "Variation loss". They perform an empirical study on traffic (METR-LA, PEMS-Bay), trust (Bitcoin-OTC, Bitcoin-Alpha), and brain fMRI datasets. Lastly, there are empirical experiments done to conclude the superior performance of DynaSTy in contrast to other strong STGNN baselines like DCRNN, Graph WaveNet, and MTGNN

**Strengths:**

S1. The paper is easy to follows.

S2. The paper addresses a major limitation of most existing Spatiotemporal Graph Neural Networks (STGNNs) like DCRNN, STGCN, and MTGNN, which assume a single, static adjacency matrix for all time steps and sample.

**Weaknesses:**

W1. Fixed Node Set Assumption: The most significant limitation, explicitly stated by the authors, is that the model "assumes that the node set remains fixed over time". This is "unrealistic" for many real-world dynamic graphs (e.g., social or biological networks) where nodes constantly appear and disappear.

W2. Unclear Justification for Static Graph Performance: On the traffic datasets (METR-LA and PEMS-Bay), the graph is static. DynaSTy's core novelty which is handling a dynamic $A_t$ is not actually leveraged here, yet it still achieves top performance. This suggests its strong results on these datasets are due to other components (e.g., the transformer architecture, pretraining, or loss function) rather than its primary advertised contribution.

W3. Confusing Runtime Results: The paper's text describes the $O(N^2)$ complexity as a "bottleneck", but the experimental results in Table 4 show DynaSTy has a faster average training time per epoch than most major baselines, including DCRNN, Graph WaveNet, MTGNN, and DGCRN16. This surprising and positive result seems to contradict the scalability limitations discussed in the text and is not explained.

W4. Paper presentation. The figures font size is sometimes small and not clear enough. The table size and figure size is also inconsistent.

**Questions:**

Please refer to the weaknesses section.

---

> ### Author Response · Authors · 2025-11-20
> **Response to concerns raised by Reviewer nUFV**
>
> We thank the reviewers for their constructive suggestions to improve the paper, and address the concerns below:
>
> 1. Fixed node set assumption: We agree this is a meaningful limitation, which we explicitly acknowledge in the paper. The assumption of a fixed node set is standard across nearly all STGNN baselines we compare against, including DCRNN, Graph WaveNet, STGCN, DGCRN, MTGNN, and attention-based ST-transformer models. Our intention was not to claim that fixed-node dynamic graphs fully capture all real-world settings, but rather to operate in the same benchmarking regime as prior work to ensure comparability.
> That said, DynaSTy’s edge-bias attention mechanism is agnostic to node persistence: if padded or special “inactive-node” tokens are used, the model can naturally extend to node appearance/disappearance (similar to how sequence models handle variable-length text). We will clarify this in the final draft and include a brief discussion on extending DynaSTy to variable-node settings, which is an active direction for our group.
>
> 2. Good performance on static-graph datasets: We appreciate the opportunity to clarify this. DynaSTy indeed targets dynamic graph settings for node attribute prediction, but generalizes well in static settings like in METR-LA or PEMS-Bay. To address the reviewer’s concern directly, we point to our Results table (Table 2) that isolates the effect of the dynamic graph vs. static. When using static graphs, Dynasty performs comparable to baselines, but achieves a lift in performance when using dynamic graphs where available, i.e., in the bitcoin and brain datasets.
>
> 3. Confusing runtime results: The theoretical discussion refers to worst-case quadratic complexity in the number of nodes, which is inherent to all Transformer-based STGNNs using full self-attention. However, in our experiments BRAIN, METR-LA and PEMS-Bay have small node counts (N= 200, N=207 and N=325). Several baselines (Graph WaveNet, MTGNN, DGCRN) use recursive or diffusion-based message passing with multiple propagation steps per layer, which dominate their computational cost.
> Thus, while DynaSTy is theoretically quadratic in N, in practice, it trains comparable to, if not faster than models with iterative GNN propagation.
> We will revise the text to make this distinction explicit: the bottleneck is scaling to very large graphs, not the typical sizes of real-world graphs that we often encounter.
>
> 4. Presentation issues: Thank you for bringing this to our attention. We will fix all formatting issues, increase font sizes, and ensure consistency across figure/table layouts. We are confident that this will improve readability.

---

> > ### Author Response · Authors · 2025-11-28
> > **Update to accommodate variable node sets**
> >
> > Taking into consideration feedback from reviewers, we have extended our model to support varying node sets across timesteps. This is achieved through an indicator feature that informs the model whether a node is active at a given timestep. Among our datasets, the Bitcoin Alpha and Bitcoin OTC trust networks exhibit dynamic node sets, and with this extension we observed a modest, though not substantial performance improvement. We will include in the appendix a comparison between model performance with and without accommodating dynamic node sets.
> >
> > We would also like to reiterate the central focus of our work: predicting the future attributes of nodes that already exist in the graph. Our objective is not to predict changes in network topology itself. That is a substantially different problem requiring architectures explicitly designed to model node birth, removal, or structural evolution.
> >
> > Our long-term goal is to use DynaSTy as a component within a broader dynamic network inference pipeline. In that setting, one must infer the network structure that best supports accurate prediction of node attributes over time. This, in turn, requires a forecasting module that takes historical node attributes and graph structure as input and outputs future node attributes. Our method is designed precisely to address that forecasting component, and the extensions we have added preserve this focus while offering broader applicability.

---

### Note · Program_Chairs · 2026-01-17
**Submission Desk Rejected by Program Chairs**

The following references in this submission do not refer to real documents and/or have major errors in bibliographic information:

 H. V. Jagadish, Johannes Gehrke, Alexandros Labrinidis, Yannis Papakonstantinou, Jignesh M. Patel, Raghu Ramakrishnan, and Cyrus Shahabi. Big data and transportation engineering. In Proceedings of the 31st IEEE International Conference on Data Engineering (ICDE), pp. 1260-1264. IEEE, 2014.
Yao Liu, Jian Tang, Jie Gao, Zhenguo Wang, and Wei Yang. Masked modeling of multivariate time series with transformer. In Proceedings of the AAAI Conference on Artificial Intelligence, volume 37, pp. 7326-7334, 2023.